# eDNA surveys substantially expand known geographic and ecological niche boundaries of marine fishes

Loïc Sanchez [1,2*], Nicolas Loiseau[1], Camille Albouy[3,4], Morgane Bruno[2], Adèle Barroil[1,5], Alicia Dalongeville[6], Julie Deter[1,5], Jean-Dominique Durand[1], Nadia Faure[2], Fabian Fopp[3,4], Régis Hocdé[1], Mélissa Jaquier[3,4], Narriman S. Jiddawi[7,8], Meret Jucker[3,4], Jean-Baptiste Juhel[1,9], Kadarusman[10], Virginie Marques[3,4], Laëtitia Mathon[2], David Mouillot[1,13], Marie Orblin[1], Loïc Pellissier[3,4], Raphaël Seguin[1], Hagi Yulia Sugeha[12], Alice Valentini[6], Laure Velez[1], Indra Bayu Vimono[12], Fabien Leprieur[1], Stéphanie Manel[11,13]

1 MARBEC, Univ Montpellier, CNRS, IFREMER, IRD, Montpellier, France, 2 CEFE, Univ Montpellier, CNRS, EPHE, IRD, Montpellier, France, 3 Unit of Land Change Science, Swiss Federal Research Institute WSL, Birmensdorf, Switzerland, 4 Ecosystem and Landscape Evolution, Institute of Terrestrial Ecosystems, Department of Environmental System Science, ETH Zürich, Zürich, Switzerland, 5 Andromède océanologie, Mauguio, France, 6 SPYGEN, Le Bourget-du-Lac, France, 7 Zanzi Marine and Coastal Solutions Chukwani, Zanzibar, Tanzania, 8 Institute of Marine Sciences Zanzibar, Zanzibar, Tanzania, 9 Laboratoire Écologie, Évolution, Interactions des Systèmes Amazoniens (LEEISA), Université de Guyane, CNRS, IFREMER, Cayenne, France, 10 Politeknik Kelautan dan Perikanan Sorong, KKD-LOLITA Bioreproduksi dan Genetika, Papua Barat Daya, Indonesia, 11 CEFE, Univ Montpellier, CNRS, EPHE-PSL University, IRD, Montpellier, France, 12 Research Center for Oceanography, National Research and Innovation Agency, Ancol Timur, Jakarta Utara, Indonesia, 13 Institut Universitaire de France, Paris, France

* loic.sanchez.ls@gmail.com

## Abstract

Assessing species geographic distributions is critical to approximate their ecological niches, understand how global change may reshape their occurrence patterns, and predict their extinction risks. Yet, species records are over-aggregated across taxonomic, geographic, environmental, and anthropogenic dimensions. The under-sampling of remote locations biases the quantification of species geographic distributions and ecological niche for most species. Here, we used nearly one thousand environmental DNA (eDNA) samples across the world's oceans, including polar regions and tropical remote islands, to determine the extent to which the geographic and ecological niche ranges of marine fishes are underestimated through the lens of global occurrence records based on conventional surveys. Our eDNA surveys revealed that the known geographic ranges for 93% of species and the ecological niche ranges for 7% of species were underestimated, and contributed to filling them. We show that the probability to detect a range filling for a given species is primarily shaped by the GBIF/OBIS sampling effort in a cell, but also by the number of occurrences available for the species. Most gap fillings were achieved by addressing a methodological sampling bias, notably when eDNA facilitated the detection of small

**Data availability statement:** All relevant data are within the paper and its Supporting information files.

**Funding:** This work has been funded by the French National Research Agency, through generic call (PRCI, French-Swiss call, no. ANR-21-CE02-0032, SHIFTeDNA) and by the Swiss National Science Foundation (grant no. 205556). Obtained by S.M. on https://anr.fr/ and L.P. on https://www.snf.ch/en. J.D. obtained funding mainly from https://anr.fr/ for these projects:—eREF: eREF was a project led by Université de Montpellier in partnership with Andromède océanologie and Spygen with the support of Agence de l'eau RMC—Gombessa Lapérouse: The Lapérouse expedition is a Galaxea's initiative and was co-financed by the French State (Ministry of Overseas France, represented by the Prefect of Reunion Island), under the Convergence and Transformation Contract 2019-2022, measure 3.5.1.1 of the "Resilient Territories" section. Sponsors and patrons also participated in the financing: Vie Océane, Aquarium de La Réunion, Comité Régional d'Etudes et de Sports sous-marins de La Réunion, Inset, O Sea Bleu, Fondation du Crédit Agricole Réunion Mayotte, Fondation d'Entreprises des Mers Australes, Air France, Odyssée Créateur de voyages, Entropie. The financial partners of Andromède Océanologie and Gombessa Expéditions are: Blancpain, Ocean Commitment Blancpain, RGBlue, Keldan, Topstar, Bigblue, Ap Diving, Molecular, Subspace, Nikon, Aqualung, Seacam.—ANGE: ANGE was a project led by the University of Montpellier in partnership with Andromède océanologie and Offshore Fishing, with the support of Agence de l'eau Rhône Méditerranée Corse (financial aid agreement N° 2021 0478) and Office Français de la Biodiversité—Parc Naturel marin du Cap Corse et de l'Agriate (R&D contract N° OFB-21-0214). This project was part of Gombessa expeditions led by Andromède Océanologie and supported by Manufacture de Haute Horlogerie Suisse Blancpain and Blancpain Ocean Commitment, the Prince Albert II de Monaco Foundation, the Société des explorations de Monaco, Office Français de la Biodiversité, and Agence de l'eau Rhône-Méditerranée-Corse (French Water Agency) with the help of ARTE, Les Gens Bien Production, CNC, Ushuaïa TV, AP diving, Aqualung, Nikon, Molecular, Seacam, Yamaha,

fishes in previously sampled locations using conventional methods. Using a machine learning model, we found that a local effort of 10 eDNA samples would detect 24 additional fish species on average and a maximum of 98 species in previously unsampled tropical areas. Yet, a null model revealed that only half of ecological niche range fillings would be due to eDNA surveys, beyond a random allocation of classical sampling effort. Altogether, our results suggest that sampling in remote areas and performing eDNA surveys in over-sampled areas may both increase fish ecological niche ranges toward unexpected values with consequences in biodiversity modeling, management, and conservation.

## Introduction

Species geographic distributions or global range maps, based on species occurrences and expert knowledge, are the cornerstones of conservation strategies, such as the classification of threatened species on the IUCN Red List. Linking species geographic distributions to ecological conditions, known as species niches [1], is fundamental for anticipating the effect of multiple threats on biodiversity, ecosystem functioning and nature's contributions to people in the face of accelerating global change [2,3]. Species geographic distributions also serve as the foundation of Species Distribution Models (SDMs) [4], which help reveal the drivers underpinning both past and current species occurrences [5] and predict future species distributions under varying socioeconomic pathways [6,7] and climate change scenarios [8]. Given that anthropogenic stressors also shape global biodiversity distribution [9], local species extirpations and colonizations [10], and can be considered as dimensions of species ecological niches [11], they can no longer be ignored in SDMs [12].

The accuracy of these approaches depends on the quantity, quality, and representativeness of species occurrences, obtained from both opportunistic records and systematic surveys while sampling bias remains a pervasive issue that can hinder our perception and prediction of species–human–environment relationships [13,14]. Species geographical distributions and ecological niches are indeed notoriously incomplete due to field surveys and species occurrences often being sparse and over-aggregated in taxonomic, geographic, environmental, and anthropogenic dimensions [15]. For example, only 6.74% of the Earth has been sampled for animals [14]. Moreover, less than 2% of species recorded in international databases such as the Global Biodiversity Information Facility (GBIF) and the Ocean Biodiversity Information System (OBIS) account for more than 50% of occurrence records in most taxonomic groups [14]. Furthermore, the inherent challenges of accessing and sampling the poles, remote areas, and deep waters have always restricted our capacity to detect species in their entire ranges and niches. Meanwhile, the occurrence data aggregated in GBIF and OBIS often fail to capture the full extent of biodiversity, particularly in the tropics where half of the cryptobenthic reef fishes remain missed by conventional surveys [14,16]. Detection biases and accessibility issues contribute to a strong latitudinal, longitudinal, and vertical sampling bias, resulting in incomplete geographic

Paralenz, Bigblue, Neotek, Seaowl, Marlink, Subspace pictures, Suex and Francqueville.— Gombessa 6 & 6+ CAP CORSE : Data come from the scientific expedition GOMBESSA CAP CORSE (Combessa 6 and Gombessa 6+) led by Andromède Océanologie and supported by Manufacture de Haute Horlogerie Suisse Blancpain and Blancpain Ocean Commitment, the Prince Albert II de Monaco Foundation, Société des Explorations de Monaco, Office Français de la Biodiversité, Parc Naturel Marin du Cap Corse et de l'Agriate, Agence de l'eau Rhône-Méditerranée-Corse (French Water Agency) and National Geographic Society. The expedition was also supported by La Marine Nationale (France Navy) and Préfecture Maritime de la Méditerranée—PISCIS: PISCIS is a monitoring network (https://medtrix.fr/portfolio_page/piscis/) led by Andromède océanologie with the support of Agence de l'eau Rhône Méditerranée Corse. L.P. obtained funding for TOPtoTOP expeditions: the TOPtoTOP expedition was supported by an exploratory grant from the Swiss Polar Institute (https://swisspolar.ch/). This study was also supported by the LabCom DiagADNe (ANR-20-LCV1-0008) funded by the Agence Nationale de la Recherche (ANR). The funders had no role in study design, data collection and analysis, decision to publish, or preparation of the manuscript.

**Competing interests:** The authors have declared no competing interests exist.

**Abbreviations:** BRT, Boosted Regression Trees; eDNA, Environmental DNA; GBIF, Global Biodiversity Information Facility; NPP, Net Primary Productivity; OBIS, Ocean Biodiversity Information System; SBDO, Sea Bottom Dissolved Oxygen; SBT, Sea Bottom Temperature; SDM, species distribution model; SST, Sea Surface Temperature.

and ecological niche ranges for most species, given the environmental and anthropogenic uniqueness of the poles and the tropics [17].

Moreover, estimating accurate species geographical distributions and ecological niches depends on our detection capacity, minimizing the risk of false negatives (species being present but not detected). In the vast ocean, this becomes particularly difficult when dealing with small or elusive species [9,18]. For fish, this estimation is even more critical since they are sensitive to both environmental and anthropogenic pressures (*e.g.,* habitat degradation) [10], and play important roles in linking marine ecosystem functioning and human societies [19], as well as supporting key contributions to people [20]. Conventional monitoring methods, such as underwater visual or camera surveys, often miss sharks, rays, and cryptobenthic fishes in habitats such as coral reefs or kelp forests [21–23]. Consequently, these species may suffer from silent local extirpations in the face of global change [10] since their geographic distributions and ecological niches are partly unknown. Thus, there is an urgent need for more comprehensive and non-destructive methods able to detect species across wide environmental and anthropogenic gradients [24].

Environmental DNA (eDNA) metabarcoding has recently been incorporated into the ecologists' toolkit for accurately assessing fish occurrences [25,26]. By amplifying and sequencing DNA filtered in water samples, and comparing sequences to a genetic reference database, this non-invasive method can detect even the most elusive and rare fish species, including elasmobranchs [18] with a restricted spatial and temporal signature [27–31]. Yet, the extent to which eDNA surveys can complement classical surveys and global databases of marine fish occurrences in undersampled regions and ecological conditions to ultimately unveil unsuspected geographic and niche ranges has not been documented.

Here, we used over 6 years of eDNA surveys and nearly one thousand samples across the world's oceans, including polar regions and tropical remote islands, to reveal sampling biases in species geographic ranges, as well as in environmental and anthropogenic ranges (hereafter ecological niche ranges) of 478 species. Rather than relying on using SDMs with limited additional occurrences, we compared species geographic and ecological niche ranges with and without the new eDNA occurrences. We also applied a machine learning and a null model approach to assess the benefits of sampling new locations and using eDNA metabarcoding to fill gaps in species geographic and niche ranges.

## Results

### GBIF/OBIS and eDNA records

The analysis of 936 eDNA samples collected from 542 locations worldwide (Fig 1a) detected 1,225 fish species that showed 100% identity on the *teleo* marker, whose primers amplify the DNA of both teleost and elasmobranch species [32]. In order to only consider species with accurate assignations, and thus avoid false positives (species detected but not being present), we removed species from genera in which fewer than 95% of species had been sequenced in neighboring regions, reducing the dataset to 478 species (see Methods). The likelihood of misassigning unknown

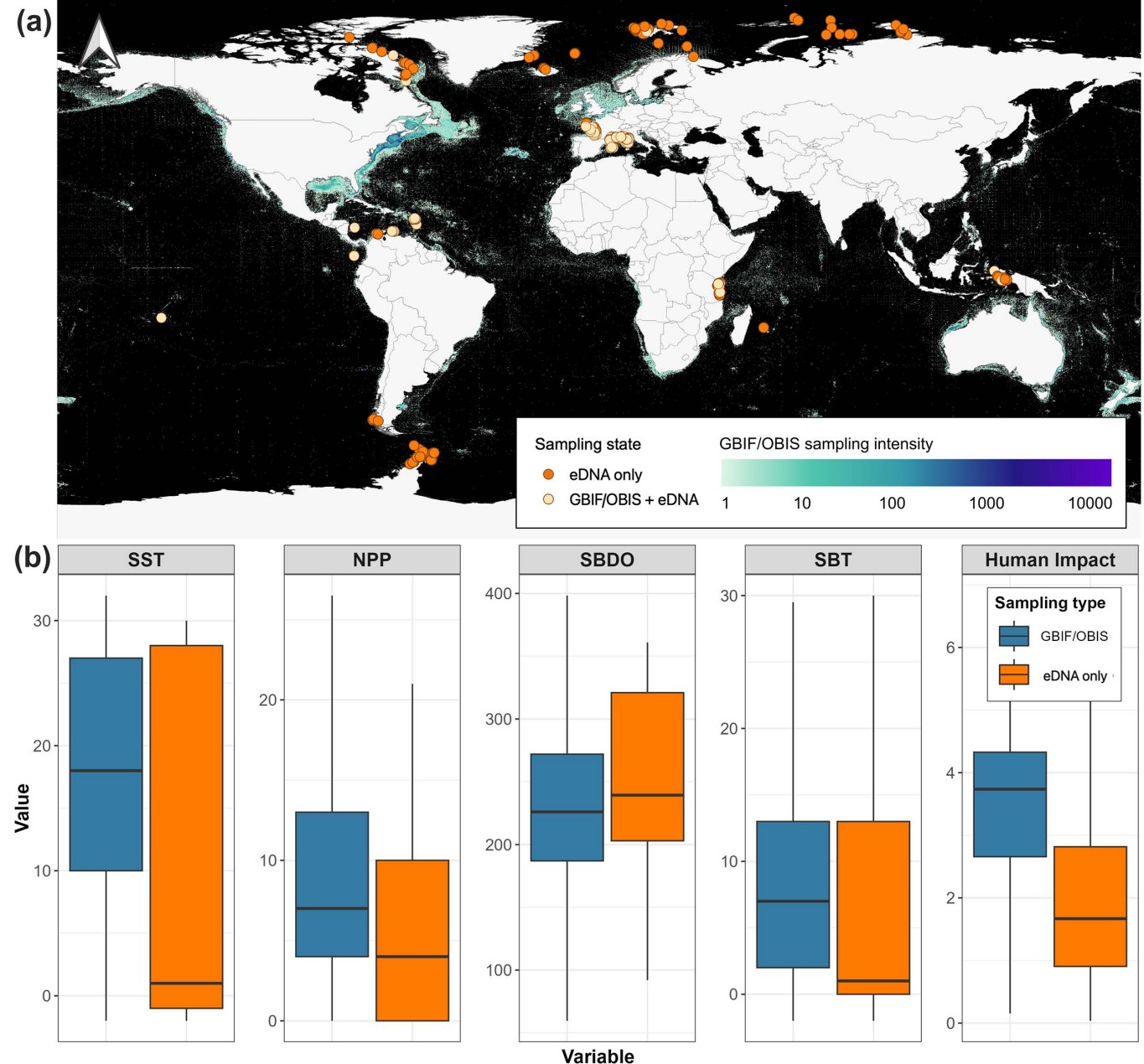

**Fig 1. Map of both GBIF/OBIS and eDNA surveys, and the distribution of their associated environmental and anthropogenic variables. (a)** Sampling intensity of GBIF/OBIS (i.e., the number of sampling events per 0.1° cell) overlaid with eDNA samples (orange dots). Cells with a dark background have not been sampled. Made with Natural Earth (public and free vector and raster map data at naturalearthdata.com). **(b)** Distributions of the five environmental and anthropogenic variables across the surveys: Sea Surface Temperature (SST; °C), Net Primary Productivity (NPP; mgC m$^{-3}$ day$^{-1}$), Sea Bottom Dissolved Oxygen (SBDO; mmol m$^{-3}$), Sea Bottom Temperature (SBT; °C), and Human Impact [28]. The values of these variables were extracted from the sampled cells, separating GBIF/OBIS-only cells (blue) and eDNA-only cells (orange). The data underlying this figure can be found in https://doi.org/10.6084/m9.figshare.30138220.

species to other genera was low, as only 463 out of 19,777 unique sequences across the entire species pool were shared between different genera (2.3%).

We extracted and downloaded occurrences from GBIF [33] and OBIS [34] databases for the 478 retained fish species, with 90% of these records dating from 1980 to 2023. Next, we gridded all eDNA and GBIF/OBIS species occurrences at 0.1° resolution (~10 km × 10 km at the equator) globally. Four types of cells can be distinguished: (i) cells with only GBIF or OBIS records (hereafter GBIF/OBIS-only), (ii) cells only sampled with eDNA (hereafter eDNA-only), (iii) cells with both GBIF/OBIS and eDNA samples (hereafter GBIF/OBIS + eDNA), and (iv) cells with no sample (hereafter unsampled). Out of the 4,335,949 global ocean cells, GBIF/OBIS fish occurrences were obtained in 223,956 cells (~5%). Our eDNA sampling effort yielded fish records in 239 cells (~0.006% of the ocean), with 110 of these cells sampled exclusively with eDNA (eDNA-only). The distribution of GBIF/OBIS occurrences reveals some vast unsampled areas, mainly in the Southern Hemisphere, the poles, the Pacific coasts of South America, and the coasts of Asia (Fig 1a). Several of the eDNA-only cells were in these previously undersampled areas (Tanzania, Lengguru in West Papua/Indonesia, the Arctic, and the Antarctic Oceans) and thus may fill gaps in our empirical knowledge of species distributions and niches.

We assigned each grid cell a value for five variables known to influence fish distributions [35] and available across the ocean: Sea Surface Temperature (SST), Sea Bottom Temperature (SBT), Sea Bottom Dissolved Oxygen (SBDO), Net Primary Productivity (NPP), and the Cumulative Human Impact Index [36]. We found significant differences in the mean value of these variables between eDNA-only and GBIF/OBIS-only cells (Fig 1b). More specifically, eDNA sampling effort covers almost the same SST range as GBIF/OBIS records but is more focused on areas with cooler SST ($t$ test = 3.78, df = 106.09, $p < 0.001$), equal SBDO ($t$ test = −1.11, df = 100.09, $p = 0.27$) and SBT ($t$ test = 0.18, df = 106.09, $p = 0.85$), lower NPP ($t$ test = 6.05, df = 101.78, $p < 0.001$), and lower Human Impact ($t$ test = 12.78, df = 110.2, $p < 0.001$).

## Benefits of eDNA surveys

In order to assess the benefits of eDNA surveys, we used Boosted Regression Trees (BRT) to model the number of species gained per cell as a function of the eDNA sampling effort, the sampling effort estimated from GBIF/OBIS, the species richness obtained from GBIF/OBIS, bathymetry, and the environmental and anthropogenic variables across the 239 eDNA sampled cells in this study. The BRT model showed very high predictive power, with a correlation of 0.96 between predicted and observed values, and a correlation of 0.83 based on a 10-fold cross-validation. Partial dependence plots highlighted that the number of species gained (over 18) plateaued after 10 eDNA samples (S1 Fig). In contrast, species gains decreased sharply with the increasing GBIF/OBIS sampling effort, reaching its minimum at around 130 GBIF/OBIS sampling records (S1 Fig). We then generated predictions with this accurate BRT model to estimate how many additional species 10 eDNA samples would yield in each cell: we found a mean gain of 23 (s.d. ± 14.52) species and a maximal gain of 98 species in undersampled, remote tropical areas (Fig 2). In our study, the cost of analyzing 10 eDNA samples corresponds to approximately 5,000 euros, excluding the cost of transport to the sampling point. So, on average, the detection cost of a new species is worth 277 euros, but is highly variable across regions.

## Geographic range filling

Aggregating all these data revealed species geographic (*i.e.,* distance between a new detection and its nearest GBIF/OBIS occurrence) and ecological niche range (the relative increase in the niche breadth) filling with our eDNA samples (Fig 3). Species range fillings detected by eDNA in cells that had previously been sampled using GBIF/OBIS address a methodological sampling bias, since the species had been missed in the surveys reporting past non-eDNA records. At the opposite, we considered that range fillings detected by eDNA in cells that had never previously been sampled address a geographic sampling bias, since we cannot quantify the extent to which eDNA facilitated the filling in these cases.

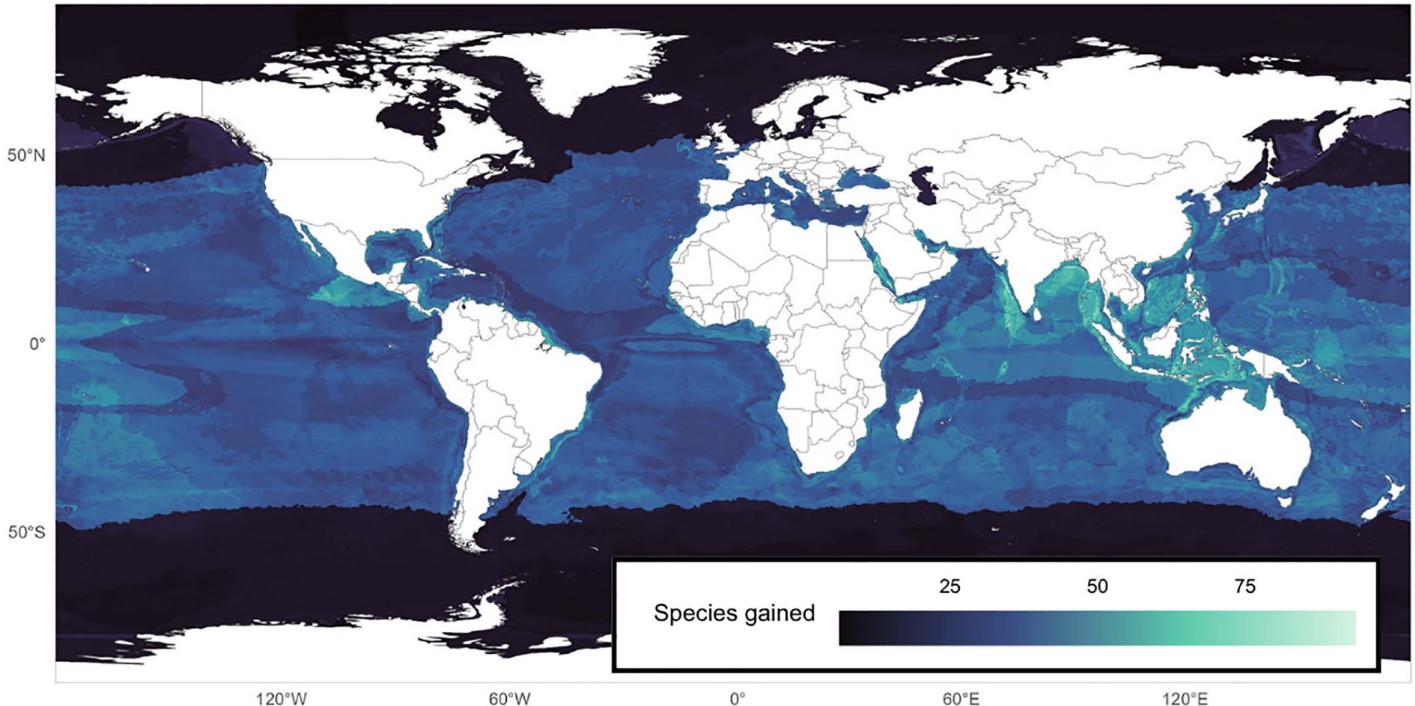

**Fig 2. Map of the predicted number of fish species gained in a cell if 10 eDNA samples were added to GBIF/OBIS sampling records, if any.** Predictions obtained from a Boosted Regression Tree using the eDNA sampling effort, the sampling effort estimated from GBIF/OBIS, the species richness obtained from GBIF/OBIS, bathymetry, Sea Surface Temperature, Net Primary Productivity, Sea Bottom Dissolved Oxygen, Sea Bottom Temperature, and the Cumulated Human Impact as predictors across the 239 eDNA sampled cells in this study. Made with Natural Earth (public and free vector and raster map data at naturalearthdata.com). The data underlying this figure can be found in https://doi.org/10.6084/m9.figshare.30138520.

For each species, we computed the geographic range filling as the longest distance by sea between each of its eDNA detections and the nearest OBIS/GBIF occurrence (Fig 3b), allowing passage through the Panama and Suez canals. If no new occurrence was detected by eDNA beyond the OBIS/GBIF geographic range, the distance was set to 0 for this species. We obtained geographic range filling for 445 out of 478 fish species, i.e., 93% of the species pool, at a maximum distance ranging from 7.4 to 10,636 km from the nearest OBIS/GBIF occurrence (S2 Fig). The mean geographic range filling for the 445 species was 472 km (±893 s.d.). The distribution of the range fillings (S2 Fig) showed that they did not result from an arbitrary choice of the cell size, since 84% of them were above 20 km, and 73% above 50 km.

We identified a total of 57 species at locations more than 1,000 km from their nearest OBIS/GBIF occurrence. Six cases of particular interest have been chosen to illustrate these exceptional geographic range fillings. A species of crocodile icefish (*Chionodraco hamatus*), that is only known in the Antarctic, was detected in Patagonia using eDNA (Fig 4a). A sampling campaign in the Falkland Islands had also detected this species [31], but GBIF classified the occurrence as "presumed swapped coordinates". Our detection confirms that this occurrence was indeed a true positive. We detected the Large-scale Mullet (*Planiliza macrolepis*) off the island of Corsica, although this species is usually found in the Indian Ocean and the Red Sea (Fig 4b), suggesting a hidden long-range colonization (>4,000 km) and a new non-indigenous fish species in the Mediterranean Sea. The Big Roughy (*Gephyroberyx japonicus*) lives at a depth of more than 300 m in temperate waters across the Western Pacific Ocean. However, a conserved specimen was found in Mauritius in 2014 [37], and we detected this species in eDNA samples in Lengguru, West Papua (3 samples), and in Tanga, Tanzania (Fig 4c) suggesting a more cosmopolitan distribution than previously thought of this species across the Indopacific. The

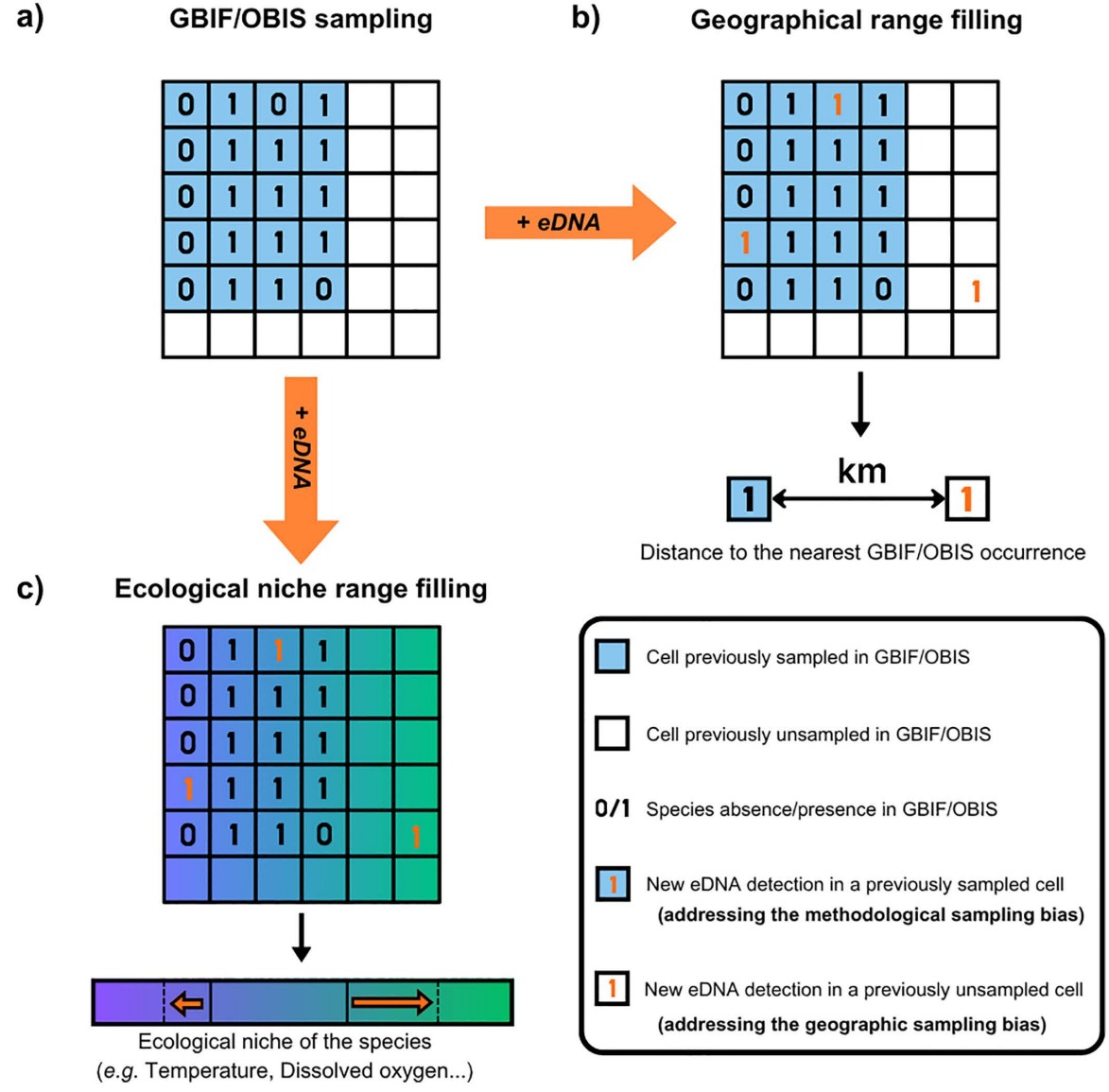

**Fig 3. Description of the framework used to analyze geographic and ecological niche range filling for each marine fish species. (a)** Example of species occurrences recorded in GBIF/OBIS on grid cells. New eDNA species detections (orange) were added to existing GBIF/OBIS occurrences (black). **(b)** Geographic range filling for a given fish species was measured as the maximum distance between new eDNA detections and the nearest GBIF/OBIS occurrence. **(c)** Ecological niche range filling for a given fish species was measured as the increase in the anthropogenic and environmental niche breadth, based on the minimum and maximum values encountered before and after updating the niche with new eDNA detections.

Phaeton Dragonet (*Synchiropus phaeton*) is usually found in the deep waters of the Eastern Atlantic Ocean. However, a Phaeton Dragonet was recorded in the Gulf of Mexico [38]. We detected this species near Malpelo, in the Eastern Pacific Ocean, Columbia, in nine different eDNA samples (Fig 4d). The only referenced congeneric species in the area that could match eDNA assignment to the reference database, the Antler Dragonet (*Synchiropus atrilabiatus*), has its complete 12S sequence available on GenBank which is different. It is therefore possible that the Phaeton Dragonet also occurs in the Gulf of Mexico, as well as in the Caribbean Sea, and may have passed the Panama Canal in an interoceanic colonization event. The Shortfin Mako (*Isurus oxyrinchus*), an endangered shark species commonly found in all temperate and

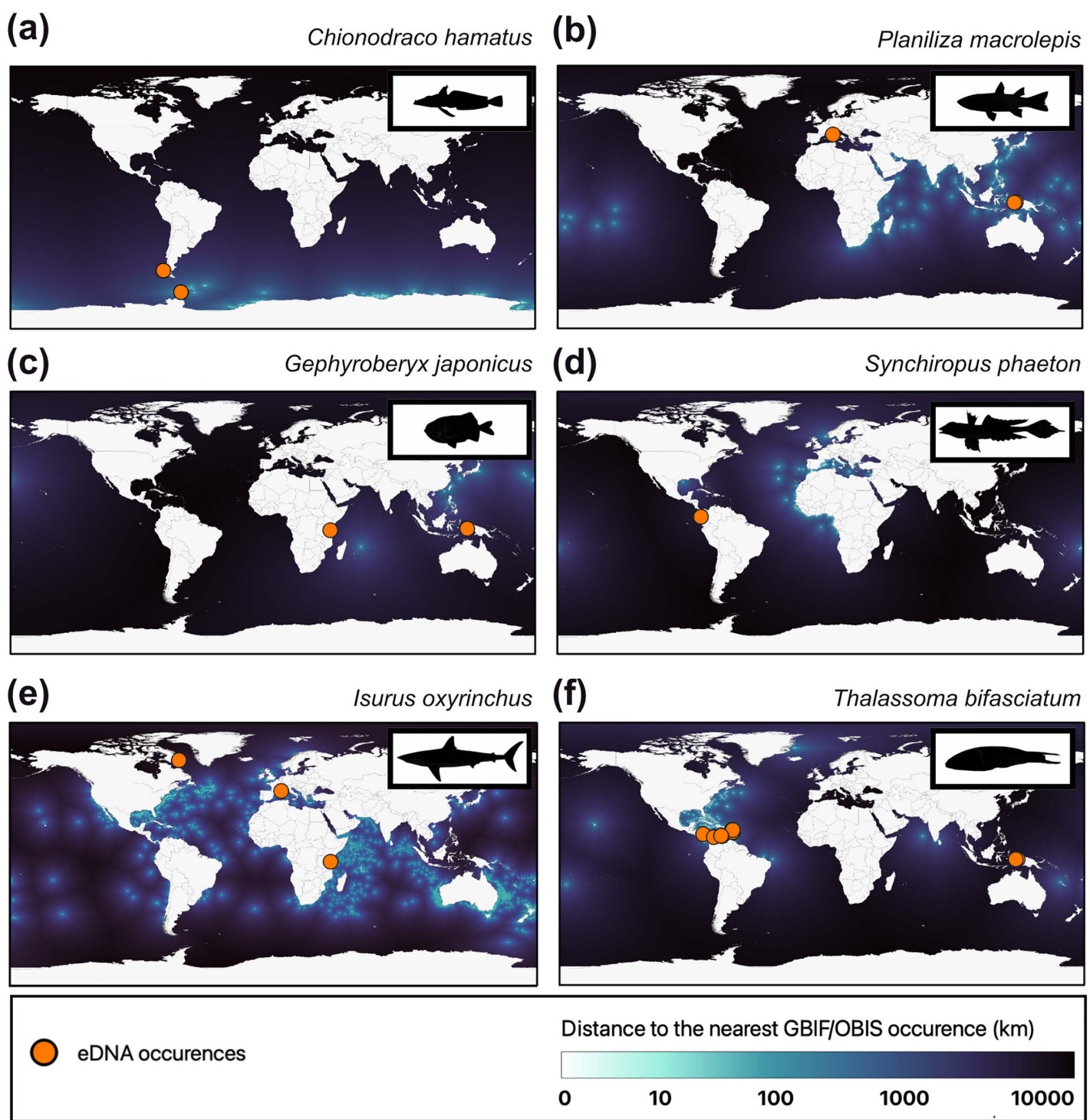

**Fig 4. Maps of the distance to the nearest GBIF/OBIS occurrence for six species detected by eDNA far from their known spatial range and qualified as geographic range filling: the darker the background, the further the cell is from a referenced GBIF/OBIS occurrence.** Detections via eDNA in our surveys are represented with orange dots. Distances are calculated by sea, and passage through the Suez and Panama canals is permitted. Made with Natural Earth (public and free vector and raster map data at naturalearthdata.com), icons made by hand. The data underlying this figure can be found in https://doi.org/10.6084/m9.figshare.30138226.

tropical seas across the world [39], was detected in our eDNA samples near the Arctic circle in the Baffin Sea (Fig 4e) suggesting a wider geographic range and niche than previously thought, thus questioning its IUCN status. Finally, the Bluehead Wrasse (*Thalassoma bifasciatum*) is usually detected in the Caribbean Sea, but five individuals were sampled and photographed in Tonga in 2006 while another study detected the species in New Caledonia through eDNA surveys in 2015 [40]. We added a detection in the Western Pacific through eDNA surveys performed in West Papua, Indonesia (Fig 4f), suggesting a wider geographic distribution across coral reefs.

## Ecological niche range filling

After associating all GBIF/OBIS fish records located in all grid cells with the corresponding environmental and anthropogenic values, we extracted all species ecological niche ranges, expressed as the range (maximum value - minimum value) of each variable for each species. Subsequently, we updated these species niche ranges using eDNA surveys and assessed the percentage of ecological niche range filling for each variable and each species, if any. We recorded a total of 66 niche fillings across 35 species (6.9% of the retained species), with 30 fillings at the lower margin of the range and 36 at the upper margin, ranging from 0.41% to 475% (Fig 5). The largest niche fillings were a 9.5 °C positive range filling in

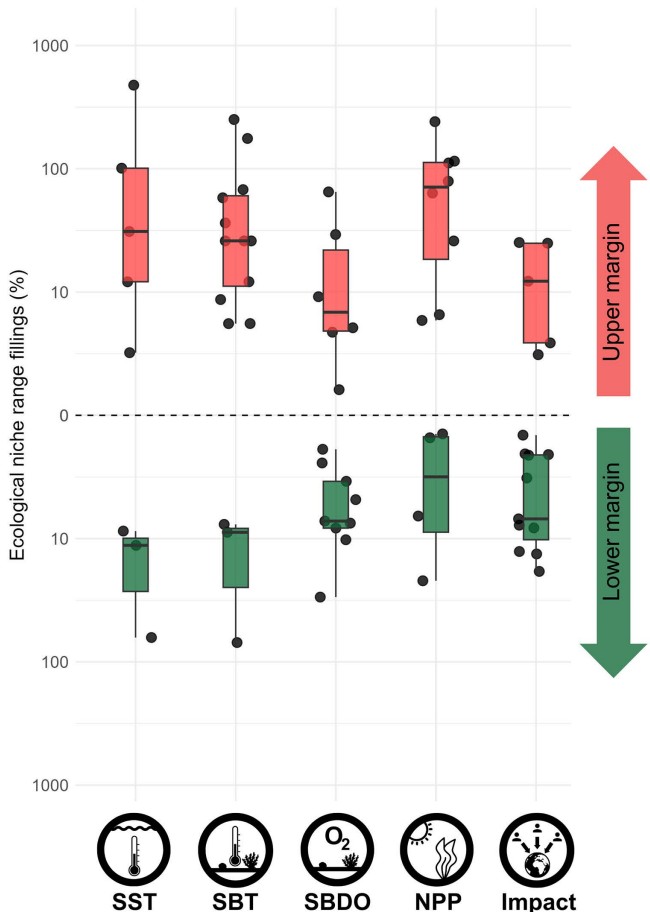

**Fig 5. Ecological niche range fillings detected with eDNA surveys for 35 fish species, calculated as the filling percentage relative to the known range, for each of the five variables.** Range fillings at the upper and lower margins of each variable are represented by black dots, and their distributions with red and green boxplots, respectively. Icons made by hand. The data underlying this figure can be found in https://doi.org/10.6084/m9.figshare.30138235.

SST for the crocodile icefish (*Chionodraco hamatus*), a 12.6 mgC m$^{-3}$ day$^{-1}$ negative range filling in NPP for *Ophiogobius jenynsi*, a 56 mmol m$^{-3}$ positive range filling in SBDO for the same species, an 11 °C negative range filling in SBT for the red-fin goby (*Evorthodus minutus*), and a 1 unit positive range filling in Human Impact for the zebra goby (*Zebrus zebrus*).

We found that 63% of these 66 ecological niche fillings were detected by eDNA surveys in cells where species were not recorded in GBIF/OBIS records. These eDNA-facilitated fillings, addressing the methodological bias (Fig 3), predominantly corresponded to cryptobenthic species (31 out of 42 niche fillings), from families such as Gobiidae, Tripterygiidae, Syngnathidae, and Apogonidae. We also found that when filling ranges by addressing the geographic bias, the maximum body length of fish species was greater than when addressing the methodological bias (Wilcoxon test, $W = 772.5$, $p < 0.001$; Fig 6).

### Drivers of geographic and ecological niche range filling

We then used two logistic regressions to quantify the probability of detecting geographic or ecological niche range fillings for any given eDNA detection, depending on species body size, total number of species occurrences in GBIF/OBIS, and GBIF/OBIS sampling intensity for the cell. For geographic range filling, the AUC of the model reached 0.94. For ecological range filling, as we have an over-representation of zeros in our dataset (10,952 out of 11,018), we randomly downsampled

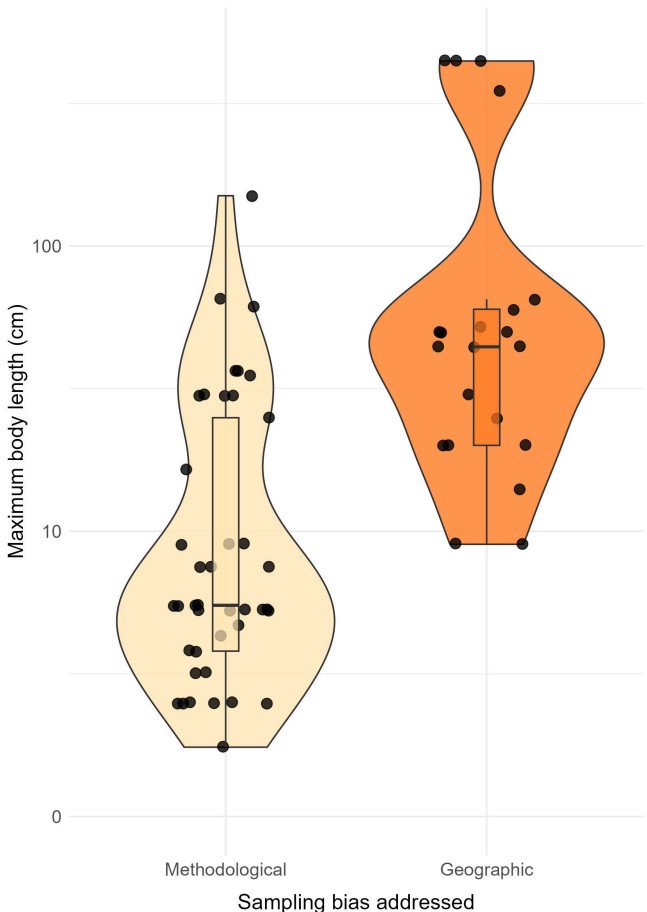

**Fig 6. Violin plots showing species maximum body length associated with each ecological niche range filling on a logarithmic scale (66 range fillings across 35 fish species).** Range fillings addressing the methodological sampling bias (cells that were previously sampled in GBIF/OBIS) and those addressing the geographic sampling bias (cells that were not previously sampled in GBIF/OBIS) are shown in light and dark orange, respectively. The data underlying this figure can be found in https://doi.org/10.6084/m9.figshare.30138244.

the dataset 1,000 times, to obtain 75% of zeros. This relationship is estimated with the median coefficient of the 1,000 models realized with randomly sampled zeros. The mean AUC across these datasets is 0.97 (s.d. ±0.007). In both models, the probability was primarily shaped by the GBIF/OBIS sampling intensity in the cell, then by the number of species occurrences in GBIF/OBIS, and finally by body size. The models show consistent trends for both types of range filling (S3 and S4 Figs). The probability of detecting a range filling is higher in unsampled cells than sampled cells (addressing the geographic sampling bias). In sampled cells, the probability depends on fish body length, as range fillings are still detected for small-bodied fish, even in highly sampled cells (addressing the methodological sampling bias). Detections of species with a lower number of occurrences have a significantly higher probability of range filling, indicating that some of the range fillings we found may only be the result of a small number of species occurrences.

We also used a null model to test whether our eDNA surveys contribute to ecological niche range filling beyond what would be expected by adding new random GBIF/OBIS occurrences (see Materials and methods). The null model indicated that nearly half (31 out of the 66) ecological niche range fillings were due to a small number of occurrences, *i.e.,* just randomly adding a number of samples, while the 35 remaining were due to new eDNA samples addressing either the geographic or the methodological sampling bias. A specific gobiid, the coralline goby (*Odondebuenia balearica*), notably had more occurrences in our new eDNA samples than reported in GBIF/OBIS, representing the archetypal situation of methodological sampling bias shadowing and shrinking the geographical and ecological niche.

## Discussion

Despite the extensive sampling efforts in recent decades, which together encompass all the world's oceans, there is still a significant bias and major gaps in the assessment of marine biodiversity based on existing global occurrence databases such as GBIF and OBIS. These biases and gaps hinder the correct estimation of species niches [13] and are critical in the context of the ongoing biodiversity crisis under accelerating global change [10]. Here, by using worldwide eDNA samples to complement GBIF/OBIS occurrences, we demonstrate geographic range filling for most of the species targeted in our study (93%) and ecological niche range filling for a few species (7%). This range filling of various magnitudes reveals geographic (*e.g.,* species detected in new cells), ecological niche (*e.g.,* species found at unexpected values of their environmental and anthropogenic boundaries), and taxonomic (*e.g.,* cryptobenthic fishes exhibited more niche range filling) biases. Most of these range fillings address a methodological sampling bias, occurring when eDNA detects a species where it was not previously recorded in GBIF/OBIS, despite the location having already been sampled. Simulations demonstrate that an average 23 species could be gained in a cell for 10 eDNA samples. Those gains would be higher in cells without GBIF/OBIS occurrences.

Species geographic range fillings were detected in all directions: poleward, equatorward, and longitudinally. This multidimensional range filling was mostly expected for rare fishes because detecting new eDNA occurrences for these species increases the likelihood of finding them far from previous records without eDNA, given that the number of records for such species is essentially small. Surprisingly, we also detected geographic range filling for species with a large number of occurrences, such as the shortfin mako shark (*Isurus oxyrinchus*), indicating that even data-rich species may have been missed by OBIS/GBIS records, specifically in undersampled areas such as the polar regions. The high proportion of species with geographic range filling (93%) corresponds to new occurrences increasing their area of occupancy by at least one cell (*i.e.,* ~10 km) for nearly all studied species. This does not necessarily reflect an increase in these species' extent of occurrence since geographic range filling of a few kilometers most likely corresponds to gap filling in the true area of occupancy (which was certainly underestimated in previous OBIS/GBIF records) or to spatial mismatch due to the fine scale of our study. However, we show that this effect is certainly marginal since most species display range filling at a coarser scale. Indeed, the large number of species found at hundreds or more than a thousand kilometers from the nearest GBIF/OBIS occurrence more likely reflects major underestimations in species' extents of occurrence suggesting that species biogeography in the ocean remains poorly understood and that more locations should be investigated with new

technologies [41]. This lack of knowledge in unsampled regions has already been highlighted for deep sea species in OBIS [41], and for all species in GBIF since 79% of GBIF data come from only 10 countries, and 37% from the USA [14]. The OBIS community acknowledges these widespread gaps across taxonomy and space [42]. Yet, no study has investigated the knowledge gaps at the species level, considering ecological niche ranges with an alternative sampling approach.

The BRT model revealed that eDNA surveys can potentially increase the number of species detected in tropical regions and that this increase depends on the number of sampling events per cell, with diminishing gains after more than one hundred GBIF/OBIS records. This result shows that in very diverse and unexplored areas, eDNA can be a valuable tool for estimating marine fish diversity. However, while a small number of eDNA samples may capture a substantial portion of the species pool, recent studies suggest that eDNA and conventional methods often detect different species subsets [43]. This underscores the value of integrating multiple sampling methods to obtain more accurate estimates of marine biodiversity and propose conservation strategies, particularly in tropical regions [25]. In polar regions, the predicted number of species gains remains quite low, likely due to incomplete genetic databases [44] and lower diversity. This highlights the urgent need to increase research efforts in these rapidly changing areas [45], which are home to unique fauna [46] due to recent speciation rates [47].

Our eDNA sampling efforts in remote, previously unsampled locations broaden niche conditions beyond those which have been extensively sampled in recent decades, adding unexpected species in some geographical areas and ecological conditions. With a rather limited number of sampled cells ($n=239$) compared to GBIF/OBIS ($n=223,956$), and a majority of coastal species detected by our eDNA sampling design [9], we filled the geographic and ecological niche ranges of all fish kinds, from large sharks with circumpolar distributions to small endemic and cryptobenthic gobies. For example, the shortfin mako (*Isurus oxyrinchus*) was detected in an eDNA-only sampled cell, 3 °C lower in SST than previously recorded, resulting in a 10% filling of its lower thermal niche margin. This finding suggests unseen poleward range shifts and represents crucial knowledge to efficiently preserve such threatened species, especially since fast range shifters often present declining population dynamics [48]. On the other side of the fish size spectrum, we detected the Liechtenstein's goby (*Corcyrogobius liechtensteini*), which is endemic to the Mediterranean Sea, by addressing the methodological sampling bias in the Mediterranean, therefore increasing its NPP range by 110%. The detections of the shortfin mako shark and of the Liechtenstein's goby illustrate that sampling in remote areas and performing eDNA surveys in over-sampled areas may both increase fish ecological niche ranges toward unexpected values. Moreover, since locations with particularly low human impact or high temperature also exist in remote tropical areas such as in West Papua or in Tanzania, we detected range fillings at both low and high niche margins, highlighting the importance of tropical samples in addition to polar ones, since future range projections from climate change could be biased due to uncharted species territories or niches [35,49]. Our results relating to temperature and oxygen are crucial since marine fishes are mostly ectotherms and may shift their ranges faster than terrestrial species [35]: a 10% filling in a fish species' thermal range is a key parameter to set up effective conservation strategies and fisheries resource management in the face of climate change. They are also of paramount importance regarding the only human-related variable included in the study: the anthropogenic niche remains poorly quantified for most species, although key in modeling and management [12].

The null model approach allowed us to quantify which ecological niche range fillings may simply have been caused by a small number of species occurrences. It reveals that half of them were significantly higher than expected under the random allocation of sampling effort. Nevertheless, this range filling still indicates that these species lack records in global databases. The main limitation of this model is that it only generates random occurrences within previously sampled locations. Since some eDNA samples occur in locations where no GBIF/OBIS records were available, the model cannot generate random occurrences in these unsampled locations. As a result, when eDNA contributes to range filling in previously unsampled regions, we cannot determine whether this effect is due to eDNA's unique ability to detect species or simply because we sampled a new overlooked location. Assessing the contribution of eDNA surveys in depth would require a null model that spatially mimics our sampling effort. However, this was not achievable because our sampling was deliberately focused in remote, unsampled areas such as the poles or Patagonia. A null model would randomly sample species

in locations where they cannot occur, thereby artificially inflating the likelihood of attributing our range fillings to random events. Alternative approaches would be to either obtain samples through conventional methods in locations where only eDNA surveys were performed, or increase the number of eDNA surveys where OBIS/GBIF data is already available [50,51], thus enabling a fair comparison between eDNA and conventional methods.

Although cryptobenthic fishes are the building blocks of reef trophodynamics [21], they are challenging to detect and underestimated by OBIS/GBIF, thus hindering the assessment of their ecological niche, geographic range, and potential extinction risk [52]. In remote locations, where space, time, and resources represent major sampling constraints, eDNA facilitates the detection of cryptobenthic species, particularly in environments rich in organic matter, where other techniques may have missed them [53]. Accordingly, our results show a significant difference in maximum body length between species range fillings addressing a methodological bias and those addressing a geographic bias [54]. The logistic regressions also indicate a lower chance of detecting range fillings for larger species, notably in highly sampled cells, even if eDNA allows the detection of large elusive species such as sharks [18]. Our result suggests that fish species of all sizes can still be detected in remote undersampled locations, while mainly small-bodied fishes can be discovered from eDNA surveys in GBIF/OBIS cells owing to eDNA's superior detection capacity over conventional methods even in the same locations [55]. While eDNA appears promising in terms of gap-filling in occurrence data, some limits persist. We may have detected unknown species that have never been recorded in a given location, or known species that have unreported intraspecific variability, and that share an identical sequence with the species we report here (false positives), or even species that diverged very recently (*e.g.,* Chaetodon family). This is particularly true in hyper-diversified regions where the available genetic reference database may cover around 40%–50% of the known checklist [44]. In addition, we know the approximate error rate for GenBank (~5%) [56,57], but we cannot precisely determine an identification error rate for GBIF and OBIS at a global scale and across so many species. This implies that, on the one hand, a small fraction of our detections may be erroneous, and on the other hand, that some ecological niches or geographic ranges computed with GBIF and OBIS may be overestimated, thus hindering the capacity to detect true range fillings when adding our eDNA samples. Nonetheless, we may have also discarded several true positives since the sequencing coverage in the region was below our 95% threshold, leading to an underestimation of our range filling. In this case, we point out the importance of better sampling remote areas, of completing genetic reference databases, and advocate the use of eDNA multi-metabarcoding primers [58]. Indeed, short markers increase the likelihood of detecting rare species and degraded eDNA whereas longer fragments could allow for better resolution at the species identification step, and finer spatial resolution since they degrade more rapidly [32,55,59]. Considering that polar and tropical regions typically have low sequencing coverage [44], which is likely to induce species range underestimations, future sampling campaigns should combine multi-primer eDNA surveys whenever possible, and species sequencing to complete genetic reference databases. Some geographic or niche range fillings may also have been missed since our first eDNA samplings date from 2017, and some of eDNA detections may have been referenced in GBIF/OBIS when data were included in a scientific publication. Some of the reported range fillings could also actually correspond to species range shifts, but our framework does not allow us to separate range shifts from range fillings. Such range shifts may happen during extreme events, triggering a new colonization event, and thus need to be documented [60]. Another reason species occurrences were certainly underestimated in our eDNA surveys is that we usually sampled around 30 L of seawater in duplicates, so we may have missed a large portion of fish assemblages since our eDNA sampling effort was relatively low. This highlights that adapted sampling strategies, including more field replicates, are required if eDNA is to be used [61]. Moreover, eDNA optimizes detection capacity locally, but cannot provide exhaustive species inventories, since only a limited area and depth layer are sampled [62]. Future sampling campaigns should, where possible, conduct eDNA surveys at different depth levels (from mesophotic zone to bathypelagic and abyssopelagic zones, etc.) in addition to shallow waters [63].

Although a limited quantity of eDNA samples collected over a short time period is sufficient to demonstrate biogeographical and ecological biases and pinpoint the crucial need for more data, it does not allow for effective integration into

SDMs. For example, the shortfin mako (*Isurus oxyrinchus*) has more than 8,500 occurrences reported in GBIF/OBIS, while we detected the species only three times with eDNA, one of them corresponding to a range filling. A novel species distribution model (SDM) which simply adds some new eDNA occurrences would have little to no impact on the estimated species niche in most modeling frameworks, especially if these occurrences represent ecological outliers.

Our observed pervasive geographic and ecological niche range fillings underline the need for more comprehensive sampling surveys to enhance conservation policies, assess sensitivity to climate change and anthropogenic stressors, improve SDMs, and most importantly, fill gaps in areas of occupancy. Since current SDMs incorporate spatial factors into their predictions [64], and threatened species are partly assessed based on their population size [65], geographic range filling is just as important as ecological niche range filling for future conservation strategies.

## Materials and methods

### eDNA collection and analyses

A total of 936 eDNA samples were collected at 542 sites, distributed across 18 countries, between 2017 and 2023 and ranging from 1 to 3,000 m depth as described in [9]. In brief, four sampling methods with direct water filtration were used: *(i)* sub-surface transect sampling from a boat, *(ii)* shallow water and deep transects by divers, *(iii)* deep transects with a towed pump [62], and *(iv)* underwater sampling bags. In most cases, 30L of seawater were filtered, except in a few cases: a lower volume when using underwater sampling bags, a higher volume for some Arctic samples. Point samplings using niskin bottles were removed from the analyses since we detected some contamination from one project to another when using this method. Filtration was performed using single-use materials, and the pump never came into contact with the samples. All surfaces used for kit preparation prior to sampling were decontaminated with bleach, and collectors wore disposable gloves, changing them between samples. DNA extraction, amplification, and sequencing were conducted in separate, dedicated rooms equipped with positive air pressure, UV treatment, and frequent air renewal. All surfaces were bleached before and after each process, and the laboratory technicians wore face masks, disposable full-body suits, hair covers and two pairs of disposable gloves [32]. Negative controls were carried out during some fieldwork and in all laboratory operations. During the metabarcoding analyses, we used the *teleo* marker (forward primer:—ACACCGCCCGT-CACTCT, reverse primer:—CTTCCGGTACACTTACCATG) [27]. The PCR process was replicated 12 times per sample to easily detect errors, and maximize the probability of detecting rare sequences, following the protocol described in S1 Methods [66]. The complete metabarcoding protocol is described in [9].

The OBITools v1.2.13 software [67] was used to process the paired-end raw sequencing files. Initially, the forward and reverse reads were assembled using *illuminapairedend*, and any joined reads with low alignment quality (less than 40) were eliminated. Subsequently, *ngsfilter* was employed to demultiplex the reads and trim the primers. The *obiuniq* tool was then used to dereplicate sequences. In order to reduce sequencing noise, sequences with a read count inferior to 10 and sequences shorter than 20 base pairs were excluded using *obigrep*. To filter out sequences most likely generated by PCR or sequencing errors, those labeled as "internal " by *obiclean*, using a threshold ratio between counts of two sequence records of 0.05, were discarded. The remaining sequences were taxonomically assigned using a reference teleo database and *ecotag*, a lowest common ancestor algorithm. The database was constructed from NCBI Genbank nucleotide database (release 254) mitochondrion sequences using *in silico* PCRs with *ecoPCR* v0.5.0 and our internal *teleo* database (v.23022023). Finally, post-processing steps were performed using R v4.3.1 to filter out errors generated by index-hopping [68] and tag-jump [69]. For tag-jump, a 0.1% abundance cut-off per library for a given sequence was applied. For index-hopping, thresholds were empirically calculated per sequencing batch using experimental blanks. After the bioinformatic analysis, the only species found in the negative controls were human and/or domestic animals, *e.g.,* (chicken, cow, pig, dog, and cat). Ultimately, only sequences with 100% species identity were retained, and sequences matching more than one species were discarded.

For each detected genus, in every sampling campaign, we recovered the GAPeDNA regional species list of the sampling provinces and neighboring provinces [44,70] and computed the regional sequencing coverage as the percentage

of congeneric species that had been sequenced in either GenBank or our custom database. In order to mitigate potential false positives (*i.e.,* detections of unreferenced congeneric species sharing the same *teleo* sequence), we discarded all detections for genera that did not reach 95% regional sequencing coverage, reducing the dataset from 1,225 to 467 species. The complete fish species pool corresponds to a total of 19,777 unique sequences, out of which only 463 were shared between different genera (2.3%). The sampling areas and their neighboring provinces for each sampling campaign can be found in S1 Table. Chondrichthyan species are not referenced on GAPeDNA, so 11 species had to be manually checked and added to the dataset; their names are provided in S2 Table.

Some unrealistic detections were discarded due to suspicions of mis-identification, contamination, or human consumption for species such as *Sardinella longiceps* in the Mediterranean Sea or *Merluccius productus* near Madagascar. Finally, we discarded a detection of *Chelon ramada* in the Caribbean, since it was detected with 12 reads in a single PCR replicate, an extremely small quantity of DNA. All 11 discarded detections and their associated reasons are listed in S3 Table.

### GBIF and OBIS data

Marine fish occurrence data were extracted from two public databases: GBIF and OBIS. These databases gather occurrence data from scientific surveys using various methods, including citizen science. Only 0.4% of all OBIS records corresponded to eDNA sampling, from a citizen science program. We selected species that had been detected in our eDNA samples, and extracted the corresponding occurrences on the two databases. OBIS occurrences were downloaded using the *occurrence* function ("robis" R package, [71]), from 1,742 to 2023, with 90% of occurrences referenced after 1980. GBIF occurrences were downloaded using the *sp_occurrence* function ("geodata" R package [72]), between 2006 and 2023, for computational time reasons. All duplicates with identical spatial coordinates were removed to avoid redundancy between databases. Our analyses exclusively used species-level data.

### Environmental and anthropogenic variables

Environmental variables were extracted from a recent public dataset [73]. We gathered monthly data rasters from 2017 to 2021 and computed a single median value raster for SST, NPP, SBDO, and SBT. We also included a raster synthesizing the cumulative impacts to marine ecosystems from 19 anthropogenic stressors, including fishing, climate change, land-, and ocean-based stressors [36].

### Spatial analysis

We rasterized all GBIF/OBIS occurrence data to a 0.1° grid and counted the number of sampling events per cell, not only for the detected species with eDNA for which we extracted occurrence, but also for the 11,786 referenced species available in [74]. We considered that two species detected at the exact same coordinates represent a single sampling event. We rasterized our eDNA occurrence data on the same grid, which told us if the cell we sampled had also been sampled in GBIF/OBIS (GBIF/OBIS + eDNA cell) or not (eDNA-only cell). We then associated the corresponding SST, NPP, SBDO, SBT, and Human Impact values to each cell. In order to fill in missing values, we computed the mean of the eight neighboring cells (queen's case), which further allowed us to associate values to sample sites that were slightly displaced on land.

### Species geographic and ecological niche ranges

The geographic distance between novel eDNA and GBIF/OBIS species occurrences was quantified by computing, for each species, a raster providing each cell's centroid distance to the nearest occurrence in GBIF/OBIS data using the *gridDist* function ("terra" R package; [75]). We then computed a geodetic distance in kilometers by sea between new eDNA occurrences and GBIF/OBIS occurrences. Since some species move through the Panama and Suez canals [76], we allowed distances to be computed through both canals. We also allowed distances to be computed through the North Pole.

For each of the 478 retained species, we calculated the range of values in which each of our five variables occurred (maximum value − minimum value), using the GBIF/OBIS data, and the augmented eDNA dataset (GBIF/OBIS coupled with eDNA), allowing us to calculate unidimensional range fillings at the upper and lower margins of each variable. If a species usually occurring between 10 and 15 °C was detected in two new cells, for example, one at 9 and the other at 20 °C, it displays 100% range filling at its upper margin and 20% range filling at its lower margin. In order to quantify how much eDNA contributed to the range filling, we separately computed both the range filling in GBIF/OBIS + eDNA cells, and the range filling in eDNA-only cells.

### Boosted Regression Trees

BRT were used to model the number of species gained in cells where eDNA had been sampled, as a function of the eDNA sampling effort, the GBIF/OBIS sampling effort, the GBIF/OBIS species richness in the cell, as well as the environmental and anthropogenic variables. We used the *dismo* package [77], implementing a 10-fold cross-validation. The response variable followed a Poisson distribution. The hyperparameters were set to a tree complexity of up to three interactions, a learning rate of 0.005, and a bagging fraction of 0.75. These values were chosen to balance model complexity and performance. Partial dependence plots were used to visualize the marginal effects of the explanatory variables on the number of species gained.

### Logistic regressions

Two different logistic regressions were computed to model the probability of an eDNA detection, across species, to be (i) a geographic range filling or (ii) an ecological niche range filling, in function of the species body length, the number of GBIF/OBIS occurrences, and the GBIF/OBIS sampling intensity of the cell. When modeling the probability of detecting geographic range filling, 0 and 1 values were almost balanced, so we included the whole dataset in the model, while using the taxonomic family of the species as a random effect. However, when modeling the probability of detecting ecological niche range filling, 0 values were largely overrepresented. In order to minimize the bias in parameter estimation, we randomly downsampled the dataset 1,000 times, to obtain 75% of zeros in the downsampled datasets. The median coefficient of the 1,000 models computed with randomly sampled zeros was then used to predict relationships between the predictors and the response variable.

### Null model

We set up a null model to test if our eDNA samplings, whether they address the methodological or geographic sampling bias, contribute to ecological niche range filling beyond what would be expected from adding GBIF/OBIS occurrences. We generated 1,000 sets of occurrences per species that were randomly selected from the total number of GBIF/OBIS occurrences. More precisely, we randomly selected a subset by removing $N$ occurrences, where $N$ corresponds to the number of new eDNA occurrences for a given species. For each of these random samples, we then computed the species ecological niche range filling after adding the $N$ randomly removed GBIF/OBIS occurrences. This produced a null distribution of expected ecological niche range filling under the assumption that adding $N$ random occurrences GBIF/OBIS changes the perceived ecological niche range of a given species. We then compared the observed range fillings (calculated when adding our eDNA occurrences) to this null distribution for each species.

### Supporting information

**S1 Fig. Partial dependence plots showing the relationship between each variable included in the Boosted Regression Trees and the number of species gained.** The data underlying this figure can be found in https://doi.org/10.6084/m9.figshare.30138472.
(PNG)

**S2 Fig. Violin and boxplot of the maximum distances between eDNA samples and their nearest associated GBIF/OBIS occurrence for each of the 445 species that showed range fillings.** The data underlying this figure can be found in https://doi.org/10.6084/m9.figshare.30138277.
(JPG)

**S3 Fig. Predicted probability of a species detection being a geographical (a) or ecological (b) range filling as a function of the number of occurrences in GBIF/OBIS, while setting the sampling intensity and body length at their median value.** For ecological niche range fillings (b), as we have an over-representation of zeros in our dataset (10,952 out of 11,018), we randomly downsampled the dataset 1,000 times, to obtain 75% of zeros in the downsampled datasets. This relationship is estimated with the median coefficient of the 1,000 models realized with randomly sampled zeros. Therefore, the predicted probabilities do not represent absolute, realistic values but rather an estimated trend. The data underlying this figure can be found in https://doi.org/10.6084/m9.figshare.30146266.
(JPEG)

**S4 Fig. Predicted probability of a species detection being a geographical (a) or ecological (b) range filling as a function of the body length of the fish species in cm, while setting the number of species occurrences in GBIF/OBIS at its median value.** This relationship is represented for three different cell sampling intensities from unsampled (0 sampling events; 10th percentile), to moderately sampled (38 sampling events, 50th percentile), and highly sampled (354 sampling events, 90th percentile). As we have an over-representation of zeros in our dataset (10,952 out of 11,018), we randomly downsampled the dataset 1,000 times, to obtain 75% of zeros in the downsampled datasets. This relationship is estimated with the median coefficient of the 1,000 models realized with randomly sampled zeros. Therefore, the predicted probabilities do not represent absolute, realistic values but rather an estimated trend. The data underlying this figure can be found in https://doi.org/10.6084/m9.figshare.30146266.
(JPEG)

**S1 Table. List of the provinces used to compute the regional sequencing coverage.** Checklists of sampling areas and neighboring regions were used to compute the coverage.
(DOCX)

**S2 Table. Chondrichthyan species that were considered safe to be manually added to the dataset due to the absence of data on GAPeDNA.**
(DOCX)

**S3 Table. Table of removed species, the location they were detected at, and the associated reasons.**
(DOCX)

**S1 Methods. Supporting methods.**
(DOCX)

## Acknowledgments

Data from this paper come from the projects and expeditions listed below. We thank Laurent Ballesta, Patrick Durville, Florian Holon, and Laurent Pouyaud, for participating in the eDNA sampling.

*eREF*

Project led by the University of Montpellier and co-financed by Agence de l'eau Rhône Méditerranée Corse (French Water Agency), Andromède Océanologie and Spygen

*Gombessa Lapérouse*

The Lapérouse expedition is a Galaxea initiative and was co-financed by the French State (Ministry of Overseas France, represented by the Prefect of Reunion Island), under the Convergence and Transformation Contract 2019–2022, measure 3.5.1.1 of the "Resilient Territories" section. Sponsors and patrons also participated in the financing: Vie Océane, Aquarium de La Réunion, Comité Régional d'Etudes et de Sports sous-marins de La Réunion, Inset, O Sea Bleu, Fondation du Crédit Agricole Réunion Mayotte, Fondation d'Entreprises des Mers Australes, Air France, Odyssée Créateur de voyages, Entropie. The financial partners of Andromède Océanologie and Gombessa Expéditions are: Blancpain, Ocean Commitment Blancpain, RGBlue, Keldan, Topstar, Bigblue, Ap Diving, Molecular, Subspace, Nikon, Aqualung, Seacam.

*ANGE*

This work was funded by the ANGE project led by the University of Montpellier and co-financed by Agence de l'eau Rhône Méditerranée Corse (French Water Agency), Office Français de la Biodiversité (OFB), Parc Naturel Marin du Cap Corse et de l'Agriate (PNMCCA), Andromède Océanologie, and Bastia Offshore Fishing. Fieldwork was also part of Gombessa 6 expedition.

*Gombessa 6 et 6+ "CAP CORSE"*

Data provided by the scientific expedition Gombessa 6 and Gombessa 6+CAP CORSE led by Andromède Océanologie and supported by Manufacture de Haute Horlogerie Suisse Blancpain and Blancpain Ocean Commitment, the Prince Albert II de Monaco Foundation, the Société des explorations de Monaco, Office Français de la Biodiversité, Agence de l'eau Rhône-Méditerranée-Corse (French Water Agency), National Geographic Society, Iris Foundation and Lemarchand Foundation. The mission was also supported by La Marine Nationale (French Navy) and Préfecture Maritime de la Méditerranée. The authors also want to thank ARTE, INPP (Institut National de Plongée professionnelle), Les Gens Bien Production, CNC, Parc Naturel Marin du Cap Corse et de l'Agriate (France Relance program), Ushuaïa TV, AP diving, Aqualung, Nikon, Molecular, Seacam, Yamaha, Paralenz, Bigblue, Neotek, Seaowl, Marlink, Subspace, Suex, Dive System and Francqueville without whom this expedition would not have been possible.

*PISCIS*

Data provided from the PISCIS project led by Andromède océanologie and co-funded by Agence de l'eau Rhône Méditerranée Corse (French Water Agency) and the University of Montpellier.

*Lengguru*

We thank the National Research and Innovation Agency (BRIN) in Indonesia and IRD in France for promoting our international collaboration with the strong support of the Sorong Polytechnic of Marine and Fisheries (Politeknik KP Sorong), in West Papua, Indonesia. We thank the International Joint Laboratory 'Sentinel Laboratory of the Indonesian Marine BiodiversiTy' (IJL SELAMAT) for its support. The Lengguru fieldwork was conducted according to relevant guidelines by the government of the Republic of Indonesia and under a research permit issued by RISTEK (Indonesia) (permit no. 3179/FRP/E5/Dit.KI/IX/2017) and the relevant Indonesian government collecting permit. Fieldwork in Indonesia and laboratory activities were supported by the Lengguru 2017 Project, conducted by the French National Research Institute for Sustainable Development (IRD), National Research and Innovation Agency (BRIN) with the Research Center for Oceanography (RCO), the Politeknik Kelautan dan Perikanan Sorong, the University of Papua (UNIPA) with the help of the Institut Français in Indonesia (IFI), funding from Monaco Explorations, and corporate sponsorship from the Total Foundation and TIPCO company.

*Malpelo*

We thank Monaco Explorations for funding fieldwork and laboratory analysis, the Yersin crew for assistance with at-sea operations, the Malpelo Foundation for the coordination of the expedition and National Parks of Colombia and the Navy of Colombia for the permits.

"We thank the José Benito Vives de Andréis Marine and Coastal Research Institute Invemar for their institutional support, and to Giomar H. Borrero-Pérez and Andrea Polanco F. for coordinating the fieldwork in Colombia, performing sample collection, and managing the process of obtaining research permits."

*Shift-eDNA*

We thank the Nunavut Research Institute and all local communities for allowing us to obtain samples in the Baffin Sea, and sharing their knowledge with our teams. We also thank Eric Brossier (Vagabond) and Sébastien Roubinet (Nagalaqa) for the fieldwork, and we have a special thought for Louis Bernatchez, who helped us prepare fieldwork in Nunavut. This work was financed by the ANR International Shift-eDNA (Call CE2-2021).

*TOPtoTOP Global Climate Expedition*

We thank Dario Schwörer for his participation in the TOPtoTOP Global Climate Expedition.

*MPA poverty*

This study was also supported by the MPA-POVERTY project (ANR-19-CE03-0005).

*LabCom DiagADNe*

This study was also supported by the LabCom DiagADNe (ANR-20-LCV1-0008) funded by the Agence Nationale de la Recherche (ANR).

## Author contributions

**Conceptualization:** Nicolas Loiseau, David Mouillot, Fabien Leprieur, Stéphanie Manel.

**Data curation:** Morgane Bruno.

**Formal analysis:** Loïc Sanchez.

**Investigation:** Loïc Sanchez, Camille Albouy, Adèle Barroil, Alicia Dalongeville, Julie Deter, Jean-Dominique Durand, Nadia Faure, Fabian Fopp, Régis Hocdé, Mélissa Jaquier, Narriman S. Jiddawi, Meret Jucker, Jean-Baptiste Juhel, Kadarusman, Virginie Marques, Laëtitia Mathon, David Mouillot, Marie Orblin, Loïc Pellissier, Raphaël Seguin, Hagi Yulia Sugeha, Alice Valentini, Laure Velez, Indra Bayu Vimono, Fabien Leprieur, Stéphanie Manel.

**Methodology:** Loïc Sanchez, Nicolas Loiseau, Morgane Bruno, Alicia Dalongeville, David Mouillot, Alice Valentini, Stéphanie Manel.

**Project administration:** Loïc Sanchez, Laure Velez.

**Resources:** Laure Velez.

**Software:** Morgane Bruno.

**Supervision:** Fabien Leprieur, Stéphanie Manel.

**Validation:** David Mouillot, Fabien Leprieur, Stéphanie Manel.

**Visualization:** Loïc Sanchez, Marie Orblin, Raphaël Seguin.

**Writing – original draft:** Loïc Sanchez.

**Writing – review & editing:** Loïc Sanchez, Nicolas Loiseau, Camille Albouy, Morgane Bruno, Adèle Barroil, Alicia Dalongeville, Julie Deter, Jean-Dominique Durand, Nadia Faure, Fabian Fopp, Régis Hocdé, Mélissa Jaquier, Narriman S. Jiddawi, Meret Jucker, Jean-Baptiste Juhel, Kadarusman, Virginie Marques, Laëtitia Mathon, David Mouillot, Marie Orblin, Loïc Pellissier, Raphaël Seguin, Hagi Yulia Sugeha, Alice Valentini, Laure Velez, Indra Bayu Vimono, Fabien Leprieur, Stéphanie Manel.

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
