## [Editor Report · Decision Letter 0]

28 Mar 2025

Dear Dr Sanchez,

Thank you for submitting your revised manuscript entitled "The uncharted geographic and ecological niche boundaries of marine fishes" for consideration by PLOS Biology.

Your revisions have now been evaluated by the PLOS Biology editorial staff, and I'm writing to let you know that we would like to send your submission out for re-review.

IMPORTANT: Please can you provide a "track changes" marked up version of your manuscript so that the reviewers can see the changes that you have incorporated? Please include this file when you upload your additional metadata (see next paragraph).

However, before we can send your manuscript back to the reviewers, we need you to complete your submission by providing the metadata that is required for full assessment. To this end, please login to Editorial Manager where you will find the paper in the 'Submissions Needing Revisions' folder on your homepage. Please click 'Revise Submission' from the Action Links and complete all additional questions in the submission questionnaire.

Once your full submission is complete, your paper will undergo a series of checks in preparation for re-review. After your manuscript has passed the checks it will be sent out for review. To provide the metadata for your submission, please Login to Editorial Manager (https://www.editorialmanager.com/pbiology) within two working days, i.e. by Apr 01 2025 11:59PM.

Kind regards,

Roli Roberts

Roland Roberts, PhD

Senior Editor

PLOS Biology

rroberts@plos.org

---

## [Decision Letter · Decision Letter 1]

27 May 2025

Dear Dr Sanchez,

Thank you for your patience while we considered your revised manuscript "The uncharted geographic and ecological niche boundaries of marine fishes" for publication as a Research Article at PLOS Biology. Your revised study has been evaluated by the PLOS Biology editors, the Academic Editor and two of the original reviewers.

Reviewer #1 re-iterates why he likes the paper, but says “I still feel the study falls a bit short in using the new eDNA data to say some more quantitat[iv]e about how to select sites for sampling and what method to use.” Essentially he doesn’t buy your argument that a cost-benefit analysis is beyond the scope of the paper, and it pushing (again) for you to do this. He also suggests a different null model. Reviewer #3 says again that you have misrepresented the study, in trying to give the impression that you used SDM.

I discussed these comments with the Academic Editor, who agreed with reviewer #1's continuing concerns. We would strongly encourage to attend to this reviewer's central concern, preferably by new analyses.

In light of the reviews, which you will find at the end of this email, we would like to invite you to revise the work to thoroughly address the reviewers' reports.

Given the extent of revision needed, we cannot make a decision about publication until we have seen the revised manuscript and your response to the reviewers' comments. Your revised manuscript is likely to be sent for further evaluation by all or a subset of the reviewers.

**IMPORTANT - SUBMITTING YOUR REVISION**

*Re-submission Checklist*

*Published Peer Review*

*PLOS Data Policy*

*Blot and Gel Data Policy*

Sincerely,

Roli Roberts

Roland Roberts, PhD

Senior Editor

PLOS Biology

rroberts@plos.org

REVIEWERS' COMMENTS:

Reviewer #1:

[identifies himself as Jonathan Belmaker]

This study uses a comprehensive and impressive global eDNA sampling campaign to show that the current environmental and geographical niche, as estimated by GBIF and OBIS, is underestimated for a vast majority of species. The actual results are not surprising, as biases in such GBIF and OBIS occurrences records are well documented. However, the sheer scale of the analyses presented here make this a strong case for the need for more, and better, sampling.

I commented on the original version of the manuscript and find that most of my comments were addressed to a certain degree. I like the use of the logistic regression and null model to try and gain deeper insights. However, I still feel the study falls a bit short in using the new eDNA data to say some more quantitate about how to select sites for sampling and what method to use. I think some more practical advice can go a long way. The authors replied to my previous comment by writing that estimating the time and funding efforts between methods seems out of the scope and could be the goal of a separate paper. I agree to some extent, but without something more concrete the take-home message is just saying we are underdamping nature and that our samples are biased in multiple ways (this is most evident in the end of the abstract). However, we already know that and as I reader I would be much happier to feel I have learned something new.

I also have a major comment on the implementation of the null model. The null model is based on randomly removed GBIF/OBIS occurrences. However, as these are aggregated in space they will a-priori represent a limited geographic and environmental space compared to the eDNA samples. A more revealing null model will be one that mimics the geographically dispersed sampling of the eDNA data. This may require some thought.

Minutia:

The second sentence of the abstract is just too long

Line 34: Instead of "whether (the answer is trivially yes) ask to what extent / where.

Line 35: "We completed known geographic ranges" . It is not clear what you mean here before reading the rest of the ms.

Lines 38-40: This is a trivial statement. Can a more original conclusion be reached?

Line 54: "without assessing their spatiotemporal congruence" - not clear what you are trying to say here.

Lines 55-58: This sentence on anthropogenic factors and SDMs is disjunct and not really relevant to the topic at hand.

Line 66: explain what GBIF and OBIS are.

Lines 73-76: This sentence is unclear. We are dealing with the difficulty of detecting small species in the open ocean. How are these issues (dispersal, key support to people) relevant?

Lines 137-138: Geographic and ecological niche range completions need a definition / explanation.

Lines 231-233 and 300-304: "some of the completions we found may only be the result of a sampling effect." - all completions are a results of limited sampling. By definition. What you mean is explained later on, but here it is not clear.

Reviewer #3:

The novelty in this study is, that by using a very impressive copllection of DNA samples of fish from around the world, the authors show how DNA can pick up species not sampled by other methods and show several species are more widesrpread than previously reported. A more in-depth examination of these range extensions would be worthwhile and of great interest to fish biologists and biogeographers.

Unfortunately, the authors try to spin the story into something it is not. In the first version they claimed they discovered range expansions when they did not; they only reported new locations for sopecies. Now they claim their work "completes" knowledge of species ranges. This is similarly incorrect. There is no doubt that more data will continue to fill gaps in our knowledge of species distributions. Thus the paper based on its present conclusions and claims has to be rejected.

Going forward, the authors have a valuable data set and insights and should focus on those. If time allowed they could match how well published records in OBIS and GBIF fill the predicted ranges of species available in AquaMAps, IUCN Red List, OBIS (has a new range map platform) or AquaX. They they could say with more confidence how their data has "completed" gaps in species geographic ranges. The way the first version was written msled referees into thinking the authors had used SDM, but now it is clear that they did not.

It is common knowledge that all sampled data are biased and geographic distributions incomplete. To claim this as a key finding is very weak and undermines the main findings of the study.

It is not clear what data they used from OBIS and GBIF, not only because they do not list the datasets used, but they seem to suggest they only used data from samples collected from 2017-2021 (stated in response to referees, or 2023 on line 370). Why? Why not use all records from an area in OBIS and GBIF. But then on lines 427 they say they downloaded OBIS data from 1742 to 2023, but GBIF only 2006 to 2023. So it is unclear what data were used for comparison. This means their whole comparison is biased because they disregarded another 100 years or more of data in this databases.

Despite a referees suggestion, the authors refuse to acknowledge that the data in OBIS and GBIF come from distinct datasets which the licence-to-use requires they be cited. A DOI does not do this. The citaiton can be very simply copied from the metadata downloaded data and added as an Appendix. Scrutiny of these datasets may well find that some come from fishery research trawls which only sample demersal fish over a certain size. Thus, those datasets will miss small gobies, etc However, scuba diving surveys may report them. The paper could then compare DNA methods with other fish sampling METHODS more truly.

The paper is also misleading in claiming to look at ecological niche when it only matches distributions to 3 environmental variables (temperature, productivity, oxygen). USing productivity is an odd choice because this does not affect the biogeographic distribution of fish species; only local and seasonal abundance. There are no ecological variables there, like species associations, habitat, predation or herbivory, etc. The use of the ocean index of human impact to define a niche is an interesting idea, but there is no justification for it. No fish distribution is restricted by human impacts (some marine mammals may be). The abundance of some fish is affected by fisheries, but even overfished species can still be common (e.g., cod).

Why the paper declines to add a comment on the relative costs of conventional sampling vs DNA (as recommended by a referee and is a good idea) is not clear. I am sure with some assumptions such a cost estimate would be possible.

Line 56 - human impacts do affect local species distributions, but for fish, not their global. Neither are human impacts part of a species ecological niche (but see above).

Line 66 - accusing OBIS and GBIF of "fail to capture the full extent of biodiversity" is misleading and unfair. There is no study or databases that could ever do this, including the present one (claiming "completeness"). Any data availablae depends on it being collected and published. Woudl the authors claim this journal or any other "fails to capture ....". ?

Line 88 - this seems to deliberately exaggerate the novelty of the study by adding so many caveats to the sentence. This may be the first "global" comparisons of deteciton of fish by DNA vs tradiotnal methods. But surely regional and local studies are comparable in detectability? Why are their findings not summarised?

Line 188 - this page is the most interesting results; finding species are more widespread is worth exploring further.

Line 275+ - we do not need analysis to tell us that DNA (or any method) is more likely to find an unrecoreded species in a place where there are no recorded species.

Line 261 - claiming the DNA + OBIS + GBIF is now a complete range, and then calculating how much DNA contributed to this "completion" is a circular self-made argument. If they plotted species discovery curves for areas then they may get an idea of completion (e.g., see paper by Mora et al in Proc. R. Soc B some years ago), but that was not done here.

Maps in Fig1 (a) and fig 3 blend dark blue into black to it seems the open ocean is the mbest sampled. Also very poor resolution.

---

## [Editor Report · Decision Letter 2]

8 Sep 2025

Dear Dr Sanchez,

Thank you for your patience while we considered your revised manuscript "The uncharted geographic and ecological niche boundaries of marine fishes" for publication as a Research Article at PLOS Biology. This revised version of your manuscript has been evaluated by the PLOS Biology editors and the Academic Editor.

Based on our Academic Editor's assessment of your revision, we are likely to accept this manuscript for publication, provided you satisfactorily address the following data and other policy-related requests.

IMPORTANT - please attend to the following:

a) Please make your Title more declarative and explicit for our broader readership. We suggest "Environmental DNA surveys substantially expand known geographic and ecological niche boundaries of marine fishes"

b) The Academic Editor said "My only concern is that I'm not sure how [reviewer 1's] second major comment is addressed. I would have liked some more text in the Discussion talking about this limitation explicitly." Please attend to this (I think this is the point about the null model?)

c) Your Financial Disclosure statement currently says “The author(s) received no specific funding for this work.” However, the Acknowledgements section of the manuscript mentions multiple sources of funding. Please complete the Financial Disclosure statement so that it accurately reflects your funding (including funders’ URLs).

d) Please address my Data Policy requests below; specifically, we need you to supply the numerical values underlying Figs 1AB, 2, 4ABCDEF, 5, 6, S1, S2, S3, S4, either as a supplementary data file or as a permanent DOI’d deposition. I note that you already have an associated Figshare deposition, but this currently only has a single very large zipped file which presumably contains the raw dataset. Please complete this deposition with the numerical values underlying the Figures.

e) Please cite the location of the data clearly in all relevant main and supplementary Figure legends, e.g. “The data underlying this Figure can be found in S1 Data” or “The data underlying this Figure can be found in https://zenodo.org/records/XXXXXXXX

f) Please make any custom code available, either as a supplementary file or as part of your data deposition.

We expect to receive your revised manuscript within two weeks.

*Published Peer Review History*

*Press*

Sincerely,

Roli Roberts

Roland Roberts, PhD

Senior Editor

rroberts@plos.org

PLOS Biology

DATA POLICY:

Regardless of the method selected, please ensure that you provide the individual numerical values that underlie the summary data displayed in the following figure panels as they are essential for readers to assess your analysis and to reproduce it: Figs 1AB, 2, 4ABCDEF, 5, 6, S1, S2, S3, S4. NOTE: the numerical data provided should include all replicates AND the way in which the plotted mean and errors were derived (it should not present only the mean/average values).

CODE POLICY

DATA NOT SHOWN?

---

## [Editor Report · Decision Letter 3]

22 Sep 2025

Dear Dr Sanchez,

Thank you for the submission of your revised Research Article "eDNA surveys substantially expand known geographic and ecological niche boundaries of marine fishes" for publication in PLOS Biology. On behalf of my colleagues and the Academic Editor, Andrew Tanentzap, I'm pleased to say that we can in principle accept your manuscript for publication, provided you address any remaining formatting and reporting issues. These will be detailed in an email you should receive within 2-3 business days from our colleagues in the journal operations team; no action is required from you until then. Please note that we will not be able to formally accept your manuscript and schedule it for publication until you have completed any requested changes.

Sincerely, 

Roli Roberts

Senior Editor

PLOS Biology

rroberts@plos.org